# Optimal sequence-based design for multi-antigen HIV-1 vaccines using minimally distant antigens

Eric Lewitus[1,2], Jennifer Hoang[1,2], Yifan Li[1,2], Hongjun Bai[1,2], Morgane Rolland[1,2]*

**1** U.S. Military HIV Research Program, Walter Reed Army Institute of Research, Silver Spring, Maryland, United States of America, **2** Henry M. Jackson Foundation for the Advancement of Military Medicine, Inc., Bethesda, Maryland, Uniiited States of America

* mrolland@hivresearch.org

## Abstract

The immense global diversity of HIV-1 is a significant obstacle to developing a safe and effective vaccine. We recently showed that infections established with multiple founder variants are associated with the development of neutralization breadth years later. We propose a novel vaccine design strategy that integrates the variability observed in acute HIV-1 infections with multiple founder variants. We developed a probabilistic model to simulate this variability, yielding a set of sequences that present the minimal diversity seen in an infection with multiple founders. We applied this model to a subtype C consensus sequence for the Envelope (Env) (used as input) and showed that the simulated Env sequences mimic the mutational landscape of an infection with multiple founder variants, including diversity at antibody epitopes. The derived set of multi-founder-variant-like, minimally distant antigens is designed to be used as a vaccine cocktail specific to a HIV-1 subtype or circulating recombinant form and is expected to promote the development of broadly neutralizing antibodies.

**Data Availability Statement:** Sequences analyzed in this study are available in GenBank under accession numbers: MN791130—MN792579, ON959609 - ON959788. The code and data

## Author summary

Diverse HIV-1 populations are generally thought to promote neutralizing responses. Current leading HIV-1 vaccine design strategies maximize the distance between antigens to attempt to cover global HIV-1 diversity or serialize immunizations to recapitulate the temporal evolution of HIV-1 during infection. To date, no vaccine has elicited broadly neutralizing antibodies. As we recently demonstrated that infection with multiple HIV-1 founder variants is predictive of neutralization breadth, we propose a novel strategy that endeavors to promote the development of broadly neutralizing antibodies by replicating the diversity of multi-founder variant acute infections. By training an HIV-1 Env consensus sequence on the diversity from acute infections with multiple founders, we derived in silico a set of minimally distant antigens that is representative of the diversity seen in a multi-founder acute infection. As the model is particular to the input sequence, it can produce antigens specific to any HIV-1 subtype or circulating recombinant form (CRF). We

generated during this study are available at https://www.hivresearch.org/publication-supplements.

**Funding:** This work was supported by a cooperative agreement between The Henry M. Jackson Foundation for the Advancement of Military Medicine, Inc., and the U.S. Department of the Army [W81XWH-18-2-0040]. The funders had no role in study design, data collection and analysis, decision to publish, or preparation of the manuscript.

**Competing interests:** I have read the journal's policy and the authors of this manuscript have the following competing interests: A patent application on invention disclosed in this publication is filed. MR and EL are the co-inventors.

applied this to HIV-1 subtype C and obtained a set of minimally distant antigens that can be used as a vaccine cocktail.

## Introduction

The diversification of HIV-1 is mediated by low replication fidelity, large population size, high recombination rates [1–3] and escape from immune pressures, including that exerted by neutralizing antibodies (nAbs) [4,5]. In the first months of infection, nAbs direct their response to recognize Envelope (Env) targets in autologous viruses [6] before heterologous targets are recognized typically a couple of years later [7,8]. While most individuals living with HIV-1 develop nAbs, only 10–25% of individuals elicit nAbs that can neutralize >70% of a diverse virus panel [9,10]. The maturation time needed to induce broadly nAbs (bnAbs) indicate that it is a complex, multi-factorial process. So far, this process has not been reproduced by vaccine candidates. Several viral factors have been associated with the development of bnAbs, including viral load, viral subtype, CD4+ T cell count, infection duration and viral diversity [11–20].

Theoretical and empirical data show that increased Env diversity in acute or early infection may prime the immune system to develop bnAbs. A modeling study by Luo and Perelson showed that broadly neutralizing responses could emerge earlier in infections founded by multiple strains as bNAbs were less likely to develop after infection with single variants due to competitive exclusion by the autologous antibody response [21]. Relatedly, the large increase in diversity that occurs following a super-infection has been linked to the subsequent development of bnAbs in these individuals [22–24]. We recently analyzed 3,482 HIV-1 Env sequences sampled from 70 people living with HIV-1 (PLWH) who were diagnosed in acute infection [25–27] and compared the sequence data to neutralization breadth measurements performed on samples collected between six months and four years after infection [28]. Participants had been enrolled in the prospective HIV-1 acute infection cohort RV217 [29]. In this cohort, more than 3,000 seronegative individuals from four countries (Kenya, Tanzania, Thailand and Uganda) were tested twice-weekly for HIV-1 RNA and 155 acute infections were identified. RV217 PLWH were followed for up to five years after viremia was detected. We showed that individuals with infections established with multiple HIV-1 founder viruses were more likely to develop neutralization breadth than those with infections established with single founder viruses [27]. This finding was reproduced in the cohort of placebo recipients who were infected during the RV144 vaccine efficacy trial [27,30,31]. We interpret recent data from small cohorts of infants living with HIV-1 as also supporting this relationship between multi-variant infections and the development of breadth. Infections with more diverse viral populations were associated with the development of neutralization breadth [32] and the proportion of infections with multiple founders was high among infants (7/12) in a different small cohort [33].

Infections with multiple HIV-1 founder variants account for approximately 25% of infections [25,34–37]. These multi-founder infections occur when multiple sequences are transmitted from a single transmitter who was likely in the chronic phase of their infection. Hence, all the sequences in the recipient are closely related, phylogenetically linked, and show around 1% intra-participant diversity (i.e., minimally distant). We previously showed that infections with multiple founder variants have clinically relevant features as viral load set point was 0.3 log10 higher in infections with multiple founders when compared to single founder infections [36]. We also recently showed that higher engagement of B cells in the first months of infection was associated with the development of bnAbs years later [28]. This finding together with the fact

that infections with multiple HIV-1 founder variants was predictive of the development of neutralization breadth [27] emphasize that the early events of HIV-1 infection play a critical role in the ontogeny of bnAbs. This led us to propose that a vaccine which would be constituted by multi-founder like, minimally distant antigens (differing by ~1% in Env) corresponding to the variability we observed in infections with multiple founder variants could initiate the induction of bnAbs. We hypothesize that a set of antigens that show minimal differences and differ primarily at surface sites, including sites that correspond to bnAb epitopes, would promote the maturation of neutralizing responses through toggle responses between variant epitopes.

An HIV-1 vaccine that elicits bnAbs is vital to prevent infection from circulating viruses. Serial administration of the bnAb VRC01 prevented HIV-1 acquisition, albeit blocking only the small fraction of circulating viruses that were highly sensitive to the bnAb (IC80 <1 μg/mL) (corresponding to 28 of 162 infections) [38]. These results emphasize the enormous challenge associated with the development of a protective HIV-1 vaccine and highlight the need for new strategies to develop vaccine candidates that could elicit neutralization breadth. Here we present a probabilistic model for simulating Env alignments that resemble a set of minimally distant antigens representative of the variability seen in multi-founder acute infections. We showed that the model recapitulated features of multi-founder variant infections, including divergence or pairwise distances across sequences and diversity at key Env antibody epitopes. Using this probabilistic model, we derived minimally distant antigens (differing by ~1%) for subtype C Env sequences and selected five sequences that can be used as a vaccine cocktail.

## Results

### A probabilistic model designed to simulate sequence alignments

The model was based on two training alignments, $F_1$ and $F_2$. $F_1$ corresponded to all the sequences descended from the major founder variant, i.e. the major founder lineage in a participant infected with multiple founder variants. $F_2$ grouped all sequences descended from minor founder variants in that participant (**Fig 1A**). Infections with multiple founder variants can be established with two founders that are clearly delineated; however, rare lineages and recombinant forms (based on the extant or unsampled founder variants) are often identified—these are sometimes found as singletons. While the model can be adapted to *n* founder lineages, for simplicity, here we considered the major founder lineage $F_1$ and grouped the other sequences in that participant (all closely phylogenetically related) as representing $F_2$. Hence, a group is not necessarily comprised of sequences descending from a unique founder variant but represents a genetically differentiable cluster with a divergent evolutionary pattern; identifying and distinguishing such clusters allow us to recapitulate the diversity of acute infections with multiple founders without biasing towards one founder variant population. The percentage of non-consensus residues at each site ($\pi_j$) was calculated for $F_1$ and $F_2$ separately, giving a mutational landscape for each founder lineage (major and minor viral population) (**Fig 1B**). Next, a transition probability matrix, $\Theta$, was computed according to the procedure described by Le & Gascuel (LG matrix) [39] for the overall rate to define the probability of each amino acid (or gap) transitioning into any other amino acid (or gap) (**Fig 1C**); the empirical transition probabilities were computed on an alignment of 172 subtype C Env sequences sampled since 2011 from the LANL HIV Database. Finally, a single sequence, aligned to the training alignment reference frame, was required to seed the model (**Fig 1D**). The single seeding sequence corresponds to the design target and can be any sequence for which a set of multi-founder like antigens is to be derived, e.g. the consensus (or most recent common ancestor) from a set of sequences corresponding to a particular participant or to a specific HIV-1 subtype

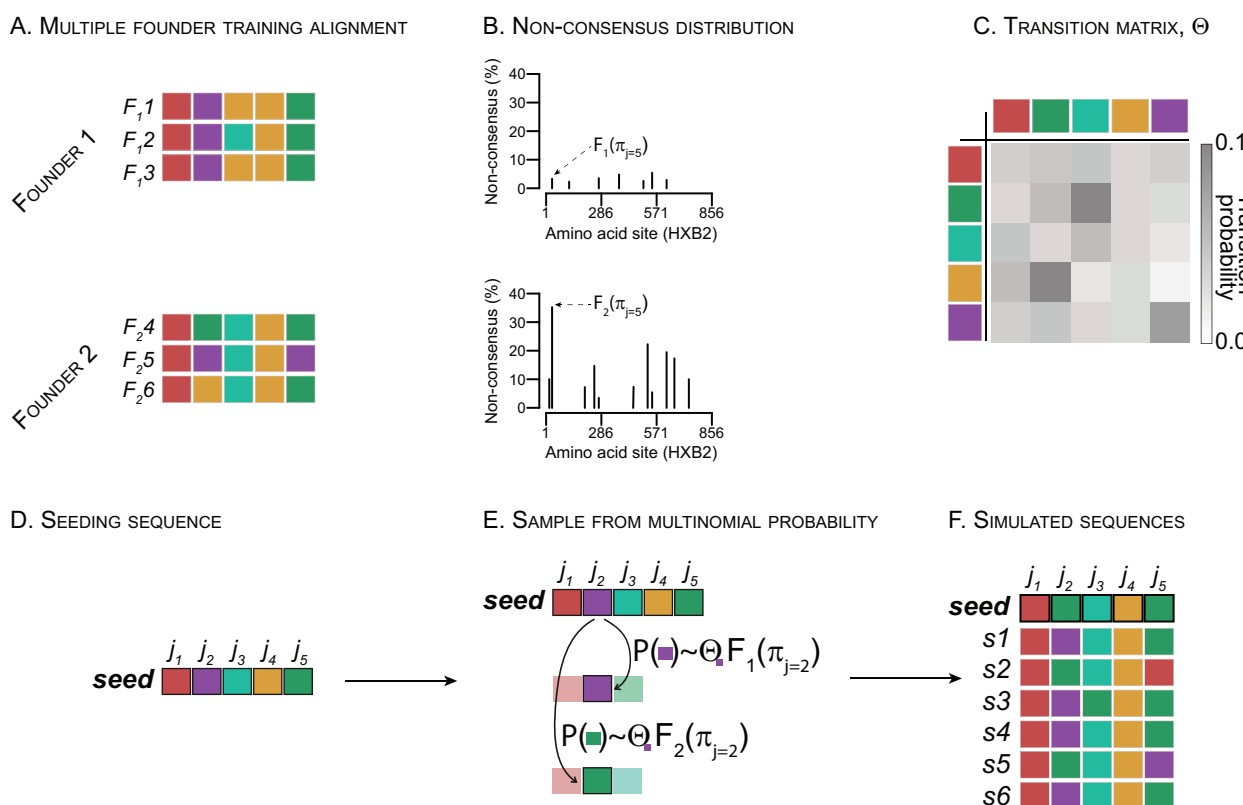

**Fig 1. Protocol for simulating alignments with multiple founder lineages.** (A) Alignments of the major ($F_1$) and minor ($F_2$) founder lineages from a participant infected with multiple founder variants are used to train the algorithm; this can also be done by pooling a set of participants in each training alignment. (B) The percentage of non-consensus amino acids is calculated at each site separately for the training alignments. (C) A transition probability matrix is calculated based on a set of empirical HIV-1 sequences. (D) A seeding sequence is input to seed the sequence simulation. (E) For each site, a residue is simulated from a multinomial probability distribution defined by the transition probability matrix and percentage of non-consensus amino acids at that site based on one of the founder variants, with an equal probability of sampling from each variant model. (F) The simulated sequences are output.

or Circulating Recombinant Form (CRF). At each iteration, *n*, sequence simulation was trained on either $F_1$ or $F_2$, such that site *j* of the seeding sequence had a probability of mutating proportional to $\pi_j$ to a new residue defined by $\Theta_j$, where $\pi$ was drawn from either $F_1$ or $F_2$ (**Fig 1E**). The model then outputs simulated sequences in an alignment (**Fig 1F**). Here we showed equal numbers of sequences derived from each founder lineage model, however, the proportion of simulated sequences trained on each founder lineages alignment can be specified. Instead of considering the variability in a given participant, the model can also rely on a pooled dataset of major vs. minor founder lineages from multiple participants. In that case, at each site, the mutational pattern can be that of any individual in the pooled set (to retain within-host diversity characteristics and forbid inter-host transitions).

## Simulated sequences reproduced the variability of acute infections with multiple founder variants

We simulated sequences trained on alignments corresponding to the major ($F_1$) and minor ($F_2$) founder lineages sampled from six RV217 participants who had infections established with multiple founder variants and whose plasma, years later, neutralized >70% of viruses on a 34-virus panel [27,28] (**Table 1**). Sequences were obtained between 4 and 34 days post-

**Table 1. RV217 infections with multiple founder variants used as training alignments.** The RV217 participant ID, Env subtype/CRF, number of sequences sampled from each founder lineage, the days post-diagnosis that sequences were sampled, and the peak neutralization breadth reached by the participant within three years of diagnosis are reported.

| ID | Subtype | Founder lineage 1 | Founder lineage 2 | Days post-diagnosis | Peak neutralization breadth (%) |
|---|---|---|---|---|---|
| 10220 | A1 | 7 | 13 | 15,31 | 74 |
| 30124 | A1 | 4 | 14 | 4,32 | 82 |
| 20337 | C | 9 | 9 | 7,34 | 79 |
| 40123 | CRF01_AE | 8 | 13 | 7,29 | 82 |
| 40363 | CRF01_AE | 10 | 10 | 7,28 | 88 |
| 40436 | CRF01_AE | 5 | 14 | 4,28 | 77 |

diagnosis. Highlighter plots (**S1 Fig**) and tree topologies (**S2 Fig**) indicated that each infection was established with multiple founder variants (**Fig 2A and 2B**). For the major founder lineage $F_1$, the mean percentage of non-consensus residues per site across the six participants was 0.06% (max = 10%) and median pairwise amino acid diversity was 0.001 (min = 0, max = 0.005). For the minor founder lineage $F_2$, mean percentage of non-consensus residues per site (1.11%, max = 50%) and median diversity (0.019, min = 0.006, max = 0.034) were both significantly higher than for $F_1$ (Mann-Whitney U test, P<0.015). An empirical transition probability matrix, $\Theta$, was computed on an alignment of 172 subtype C Env sequences sampled since 2011 and simulations were seeded with the consensus derived from sequences from an independent RV217 participant (id = 10066). One thousand sequences were simulated and trained in equal proportions on major and minor founder lineages alignments for each participant separately, where the percentage of non-consensus residues at each site, $\pi_j$, was defined by either the major founder lineages, $F_1(\pi_j)$, or minor founder lineages, $F_2(\pi_j)$. The percentage of non-consensus residues in simulated sequences at each site was highly correlated (median $R^2$ = 0.99, min = 0.98, max = 100) with the percentage in the training alignment for each founder lineage (**Fig 2C and 2D**). One thousand sequences were also simulated while trained on a pool of all founder lineage alignments across participants (**Fig 2E**), where $F_1(\pi_j)$ was the maximum percentage of non-consensus residues found at site j in a given individual across all $F_1$ alignments and $F_2(\pi_j)$ was the maximum percentage of non-consensus residues found at site j in a given individual across all $F_2$ alignments. The 95% confidence intervals of median pairwise diversity in simulated sequences included median pairwise diversity of training alignments for sequences simulated on each participant and for the pooled set of participants (**Fig 2F**). Similarly, the 95% confidence intervals of the median number of polymorphic sites per simulated sequence as well as the number of polymorphisms per sequence at CD4bs, V1-V2, V3 and MPER contact sites included the median number for training alignments for sequences simulated on each participant (**Fig 2G**).

The model was designed to simulate sequences that are specific to the seeding sequence rather than mimic the composition of the training alignment. Therefore, the simulated alignment should be genetically closer to the seeding sequence than to the training alignment. Indeed, the percentage of mismatched ungapped sites between the consensus of simulated sequences and seeding sequence (0%) was significantly lower (Mann-Whitney U test, P = 0.001) than between the consensus of simulated and consensus of the training sequences (8.27–16.12%) for sequences simulated with each training alignment (**S3 Fig**).

## Simulated sequences replicated the diversity found in infections with multiple founder variants

We compared diversity and divergence estimates for sequences simulated under the pooled set of major and minor founder lineage alignments to Env sequences sampled during acute

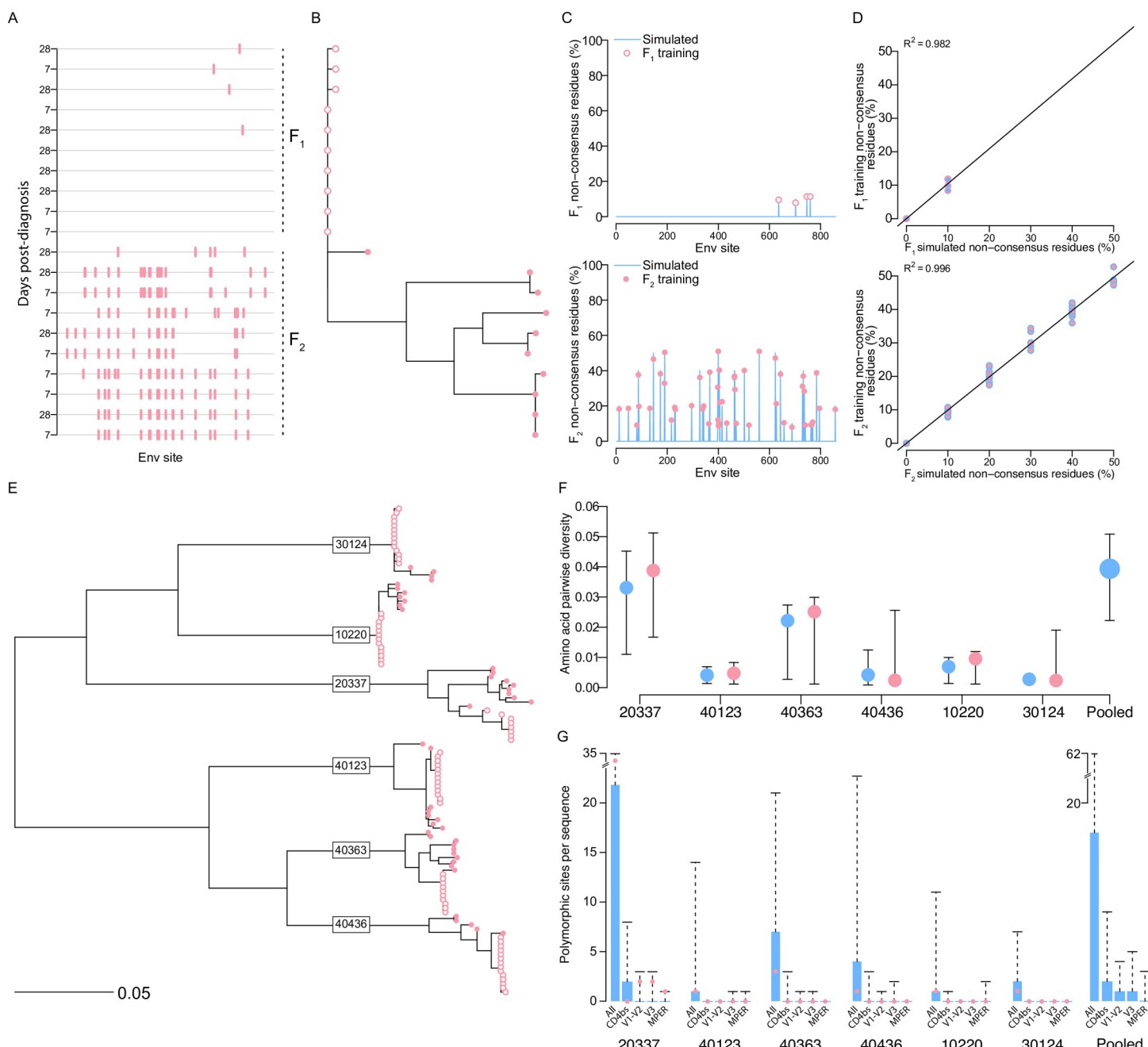

**Fig 2. Simulated sequences reproduced the variability of acute HIV-1 infections with multiple founder variants.** (A) A highlighter plot and (B) phylogeny for Env sequences sampled at 7 and 28 days post-diagnosis from a RV217 participant (id = 40363) with an infection with multiple founder variants. (C) The percentage of non-consensus residues at each Env site in sequences simulated from the major founder variant or major lineage 1 (top, blue line) and the minor founder variants grouped here as lineage 2 (bottom, blue line) after seeding with the consensus sequence from an independent acutely-infected RV217 participant (id = 10066); values for sequences belonging to the major and minor founder lineages in 40363 are shown in open and filled pink circles, respectively. (D) Regression plots of the percentage of non-consensus residues in the training alignment as a function of non-consensus residues in the simulated alignment for (top, blue fill and pink border) founder lineage 1 and (bottom, blue border and pink fill) founder lineage 2. (E) Phylogeny of sequences sampled at 4–34 days post-diagnosis from 6 RV217 participants with infections with multiple founder variants. Tips are colored to represent the population corresponding to the major (open circles) and minor (closed circles) founder populations for each participant (for simplicity, multiple founder variants or singleton sequences are grouped under the minor lineage). (F) For sequences simulated under each training alignment (see panel E), the pairwise diversity of the training alignment (pink) and of the sequences simulated under that training alignment (blue); and the pairwise diversity of sequences simulated under the pooled alignment (blue). Solid lines represent 25% and 75% interquartile ranges. (G) Barplots of the number of polymorphic sites per sequence in sequences simulated under each training alignment and the pooled-participants training alignment (blue) at all sites, CD4bs, V1-V2 contact sites, V3 contact sites, and MPER sites. Dashed whiskers indicate maximum values. Pink dots represent median values for training alignments.

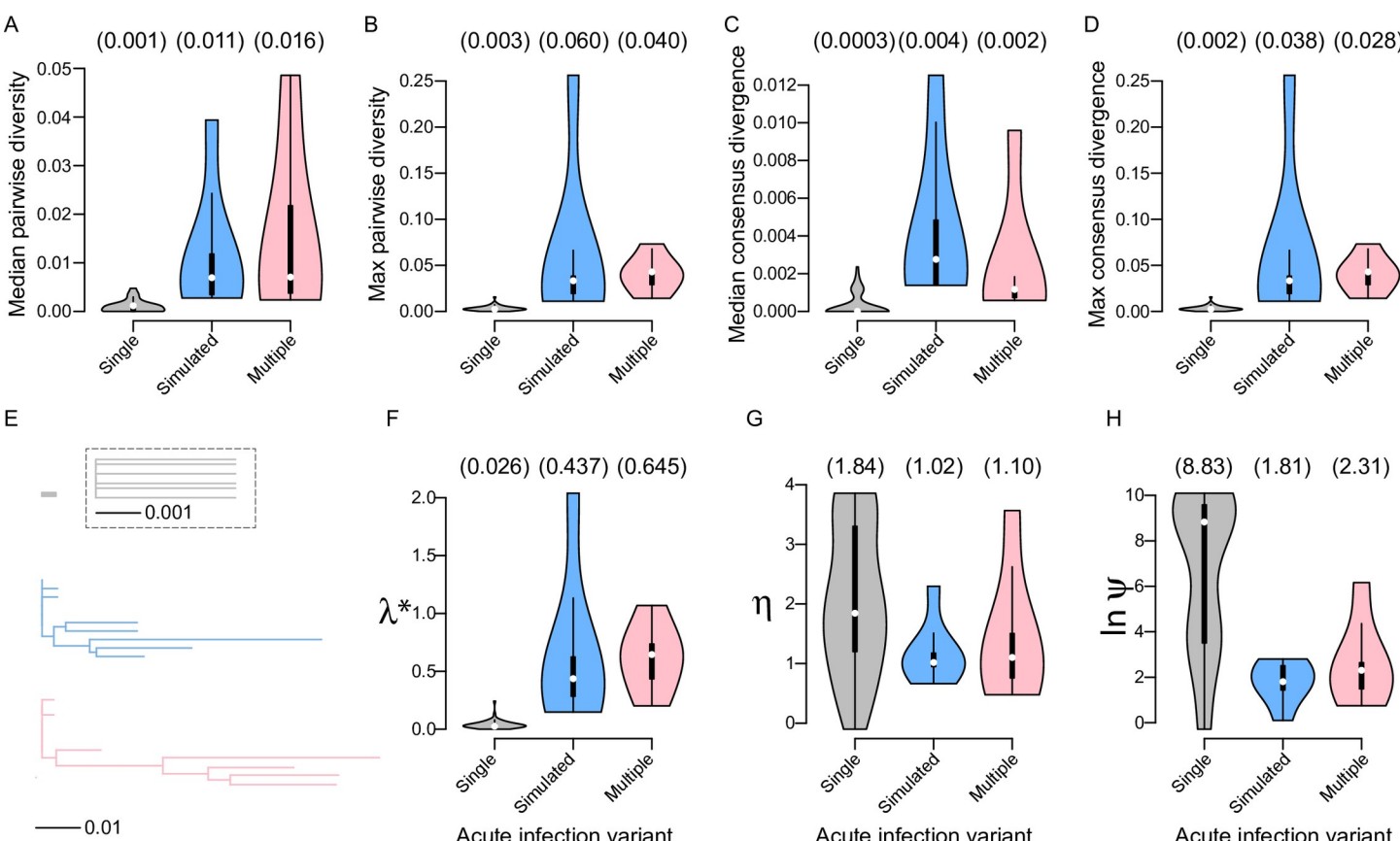

**Fig 3. Simulated sequences replicated the diversity found in infections with multiple founder variants.** Violin plots of (A) median pairwise diversity, (B) maximum pairwise diversity, (C) median divergence from the consensus, and (D) maximum divergence from the consensus for sequences sampled from RV217 participants during acute infection with a single founder variant (n = 53, grey), sequences simulated under the pooled-participants probabilistic model (n = 5000, blue), and sequences sampled from RV217 participants during acute infection with multiple founder variants (n = 6, pink). (E) Phylogenies constructed with sequences from a participant with a single founder variant sampled at 1 and 29 days post-diagnosis (id = 10066), a sample of simulated sequences from the pooled-participants model (five from major founder lineage 1 and five from minor founder lineage 2, blue), and sequences from a participant with multiple founder variants sampled at 7 and 28 days post-diagnosis (id = 40363) (pink). Phylogenies are shown at the same scale; the dashed box shows the top phylogeny at ten-times magnification. Violin plots of spectral density profile summary statistics $\lambda^*$ (F), $\eta$ (G) and ln-transformed $\psi$ (H) for the same groups. Median values are shown above each plot.

infection from RV217 participants infected with either a single founder variant (and who developed <35% neutralization breadth) (n = 12) or multiple founder variants (and who developed >70% neutralization breadth) (n = 6). The 95% confidence intervals of diversity and divergence estimates for simulated sequences included the median values for Env alignments of multi-founder variant acute infections and was higher than that of infections with single founders (**Fig 3A–3D**).

Similarly, phylogenies of simulated sequences were more similar to phylogenies of infections with multiple founder variants than to those with single founder variants (**Fig 3E**). Spectral density profiles of the modified graph Laplacian [40] were computed for Env alignments from infections with single and multiple founder variants in RV217 and for down-sampled alignments from sequences simulated under the pooled set of major and minor founder lineages alignment. Spectral density profile summary statistics each capture a unique aspect of phylogenetic topology: $\lambda^*$ is proportional to non-synonymous/synonymous rates, $\eta$ is inversely proportional to transition-transversion rate ratio, and $\psi$ is proportional to rate heterogeneity [41]. The simulated alignment was downsampled 100 times for 5 sequences simulated under each of the founder lineage training alignments, $F_1$ and $F_2$. The 95% confidence

intervals of $\lambda^*$, $\eta$, and ln-transformed $\psi$ for simulated sequences included the median values for Env alignments of acute infections with multiple founder variants but none included the median value for infections with a single founder variant (**Fig 3F–3H**).

### In silico-derived antigenic sequences for HIV-1 Env subtype C

A consensus sequence was generated from 172 subtype C Env sequences sampled after 2010 (**Fig 4A**). One thousand sequences were simulated under the pooled founders training alignment and seeded with the subtype C Env consensus sequence derived from the subtype C alignment. One-hundred samples of ten sequences (five simulated under $F_1$ and five under $F_2$) were analyzed. The sampled sequences produced phylogenies with bimodal or multimodal topologies (**Fig 4B**). The median pairwise distance of subtype C Env sequences sampled after 2010 to the consensus was 0.160 [IQR = 0.149–0.172], while the mean of median pairwise distance to the subtype C consensus across samples of simulated sequences was 0.015 (IQR = 0.013–0.016) (**Fig 4C**). Across the subtype C alignment, a mean of 15.8% of the residues at a site were non-consensus residues and 79.1% of sites (681/861) were polymorphic,

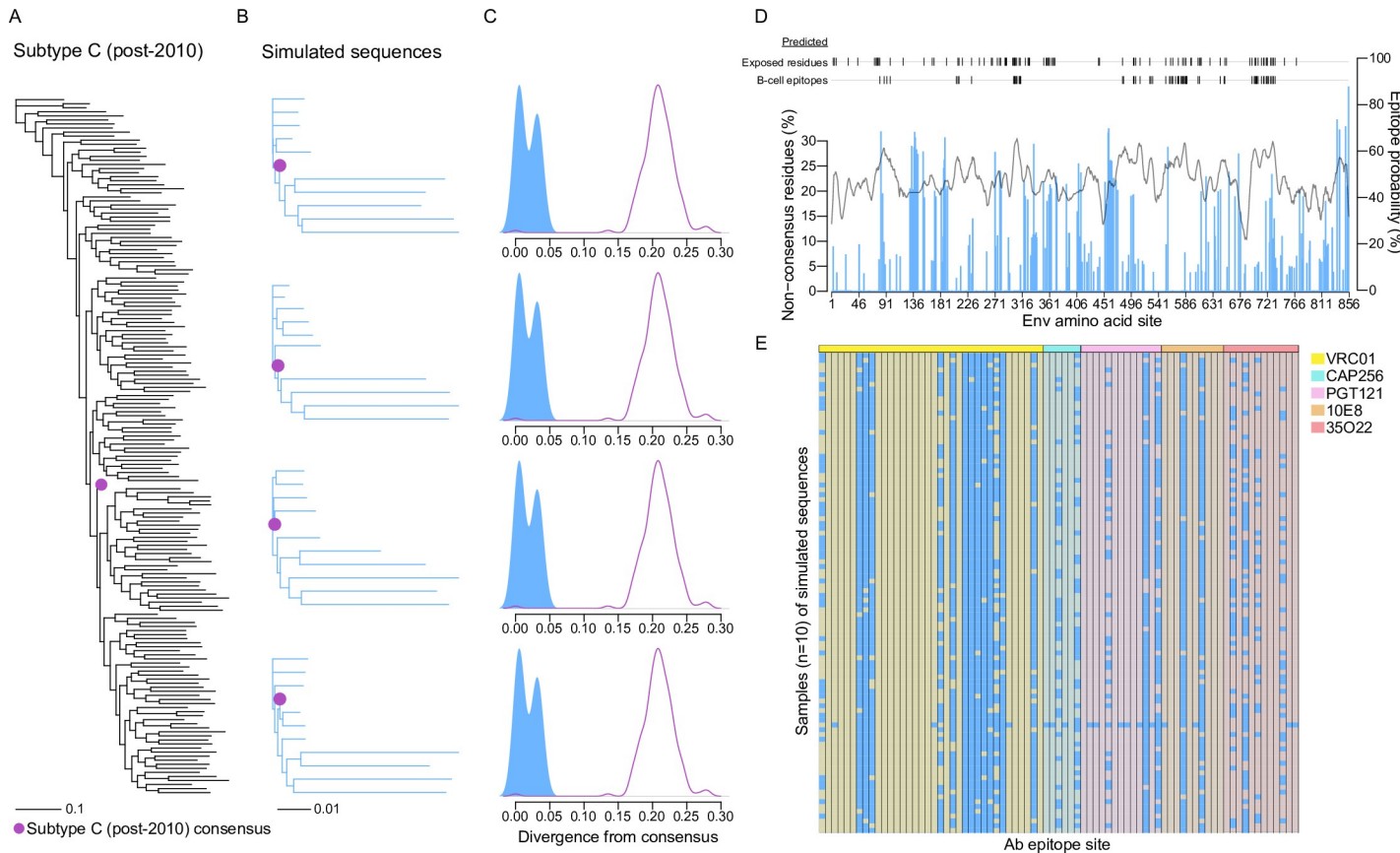

**Fig 4. Sequences simulated from the HIV-1 subtype C Env consensus.** (A) Phylogeny of subtype C Env sequences sampled after 2010. The consensus is marked in purple. (B) Four phylogenies constructed from sequences simulated under the pooled-participants model seeded with the subtype C consensus; each phylogeny is comprised of the subtype C consensus (purple) and five sequences randomly selected from simulations under major founder lineage 1 and five under minor founder lineage 2. (C) Density plots of divergence from the consensus for the alignment corresponding to each intra-host phylogeny of simulated sequences (blue) and for the inter-host subtype C phylogeny (purple). (D) Barplot of the percentage of non-consensus residues at each site in the simulated alignment. Epitope prediction probability for the subtype C consensus across sites (solid grey line). Dashes indicate the subtype C consensus predicted exposed residues and predicted B-cell epitopes that are polymorphic in the simulated alignment. (E) Polymorphisms (blue squares) at contact sites for five representative antibodies in 100 downsampled simulated alignments comprised of five sequences simulated under each founder variant lineage.

whereas across samples of simulated sequences an average (mean of median) of 3.08% [IQR = 2.93–3.16%] of residues at a site were non-consensus and 17.3% [IQR = 15.2–16%] (148/856) of sites were polymorphic (**Fig 4D**). B-cell epitopes and exposed sites were predicted for the subtype C consensus. Across samples of simulated sequences, an average of 12.3% [IQR = 11.7–13.2%] of sites (26.73/213) with >50% epitope probability were polymorphic and 16.2% [IQR = 15.0–16.9%] of sites (42.65/260) at predicted exposed sites were polymorphic (**Fig 4D**). Finally, all samples of simulated sequences were polymorphic at one or more Ab epitope sites. Five Abs corresponding to critical Env targets for neutralization were considered: VRC01:CD4bs[42], CAP256-VRC26.25:V2 apex[43], PGT121:V3[44], 10E8:MPER[45], 35O22:interface between gp120 and gp41[46]. At VRC01 epitope sites (n = 36), a median of 12 sites were polymorphic per sample of simulated sequences and 17 sites were polymorphic in at least one sample; at CAP256-VRC26.25 epitope sites (n = 6), 1 site was polymorphic per sample and 4 were polymorphic in at least one sample; at PGT121 sites (n = 13), 2 and 8; at 10E8 sites (n = 10), 2 and 2; and at 35O22 sites (n = 12), 2 and 6 (**Fig 4E**).

Finally, an alignment of simulated sequences was constructed for five Ab epitopes representative of key Env targets (VRC01, CAP256-VRC26.25, PGT121, 10E8, 35O22). There were 166 sequences with non-consensus residues in at least three of the five Ab epitopes (**Fig 5A–5F**). From these, two candidate sequences generated by $F_1$ and two by $F_2$ that were maximally divergent (within the framework of minimal diversity) were selected as candidate antigens (**Fig 5G** and **S1 File**). Together with the subtype C consensus, the candidate sequences had 20 polymorphic sites with 2–5 different residues per polymorphic site (**Fig 5H**). We predicted the structure of the simulated sequences using AlphaFold2 [47]. Predicted local distance difference tests (pLDDTs) showed generally good confidence across all domains, with median pLDDTs between 86.25–88.44, which was comparable to the median pLDDT of the subtype C Env consensus (87.77) that was used to seed the simulations. The sequences generated by $F_1$ were more similar to the seed sequence than those generated by $F_2$ (**Fig 5I**), as expected, and the structure protein prediction indicated that all simulated sequences should fold to a structured protein.

## Discussion

While bnAb infusions in humans can prevent HIV-1 infection [38], no vaccine candidate has shown the induction of such bnAbs in a vaccine efficacy trial [48–54], emphasizing the need for novel vaccine strategies. Here we developed a new vaccine design approach that emulates the diversity observed in HIV-1 infections with multiple founder variants. This multi-founder like vaccine design derives from the finding that individuals with infections established with multiple founder variants were more likely to develop bnAbs than individuals with infections established with single founder variants [27]. We showed that our design strategy reproduced the diversity seen in infections with multiple founder variants and we applied this approach to design a subtype C-specific vaccine candidate constituted of a set of five minimally distant Env sequences centered on an updated subtype C consensus.

First, we developed a probabilistic method to design antigens that reflect the diversity seen in acute infections with multiple founder variants. This method was trained on sequences descended from major and minor founder lineages sampled during acute infection from one of six individuals who developed broad neutralization breadth against HIV-1 or on a pooled alignment of sequences from all six individuals. When seeded with an independent acute infection sequence, our method generated a set of antigens that recapitulated the variability of infections with multiple founder variants, including polymorphisms at critical Env target epitopes (CD4bs, V1-V2, V3 and MPER). Importantly, the derived sequences remained close to the seeding sequence, suggesting that the simulated sequences would preserve the structural

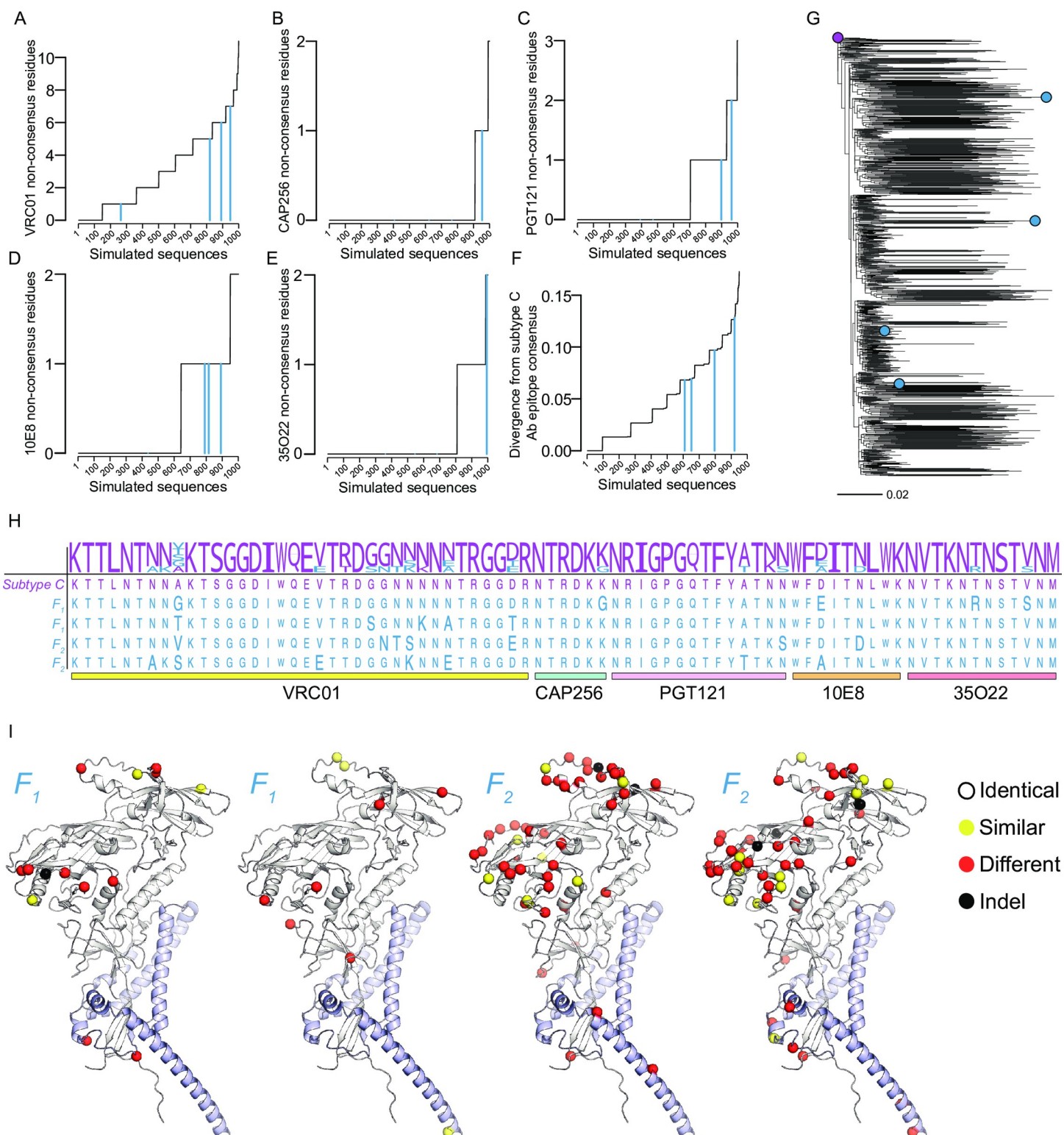

**Fig 5. In silico-derived sequences for an HIV-1 subtype C vaccine cocktail.** Line plots of the number of non-consensus residues per simulated sequence are represented for $F_1$ (dashed blue line) and $F_2$ (blue line) at epitope sites for five representative antibodies: (A) VRC01, (B) CAP256-VRC26.25, (C) PGT121, (D) 10E8, and (E) 35O22. (F) Divergence of simulated sequences from the subtype C consensus based on the five Ab epitope sites (n = 77). In A-F, sequences are sorted along the x-axis by their y-axis value; the black line traces the value across sequences. (G) Phylogeny of simulated subtype C Env sequences with tips corresponding to the subtype C consensus (purple) and candidate antigen sequences generated by $F_1$ (blue) and $F_2$ (blue). (H) Multiple sequence alignment of antibody epitope sites for the subtype C consensus and

candidate antigen sequences; non-consensus residues are shown in larger font and regions corresponding to the epitopes of VRC01, CAP256-VRC26.25, PGT121, 10E8, and 35O22 sites are highlighted below the alignment; a logo plot above the 5 candidate sequences represents the diversity found in circulating subtype C sequences at the 5 representative antibody epitopes. (I) Sequence differences from the consensus to candidate antigen sequences mapped on the predicted structure of the consensus C sequence. The $C_\alpha$ atoms of similar, different, and insertion/deletion (indel) sites are shown as yellow, red and black spheres, respectively.

integrity of the chosen central sequence. By training on alignments sampled from different infections with multiple founder variants, we showed that the variability patterns of simulated sequences differed according to the training alignment. However, for all training alignments, our method generated sequences that reflected a median genetic distance of ~1%, conforming to the infections with multiple founder variants that we want to emulate in order to elicit bNAbs.

Second, we applied this method to design a subtype C specific vaccine candidate. Seeded with the Env subtype C consensus (derived from 172 independent subtype C Env sequences sampled since 2011), we generated a set of minimally distant antigens that preserved the composition of subtype C sequences while recapitulating the variability of infections with multiple founder variants. The divergence from the consensus of simulated sequences was a magnitude smaller than that of independent subtype C sequences and so met the criteria of representing diversity around the seeding sequence while reducing the total inter-host genetic space. We selected four candidate antigens that exemplified diversity at epitope sites for representative bNAbs. However, alternative selection criteria could be used to design antigens with features that could promote other types of immune responses, such as Fc effector functions [55,56]. An important limitation of this method is its reliance on alignments from only six individuals to simulate multi-founder-like sequences. While each individual developed bnAbs, precise mechanisms behind the association between multi-founder variant diversity and the development of neutralization breadth are unknown. We tried to overcome this limitation by pooling founder lineage alignments to capture the scope of diversity found in acute infection in these broad neutralizers rather than relying on one individual to define a prototypical infection with multiple founder variants. While we chose to use as seeding sequence a subtype C consensus in order to design a subtype C vaccine candidate (because over half of the people living with HIV-1 live with subtype C), only one of the six infections we used in our pooled founder lineage alignment corresponded to subtype C (A1 (n = 2) and CRF01_AE (n = 3)). It has not been reported that patterns of variability in acute infections with multiple founders differ by subtype, nonetheless, it is possible that subtype C specific patterns exist and would not necessarily be captured by our approach. Additionally, we separated sequences into two major and minor founder lineages—this does not account for the complexity seen in some infections with multiple founder variants which can include recombinants between extant and/or unsampled sequences and rare or even unique sequences. Hence, our model simplified the landscape of acute viral diversity in these individuals. Nonetheless, if datasets of increased depths are available, the model can be used to simulate the multitude of distinct sequences that can be identified in a set of hundreds of Env sequences from acute HIV-1 infection.

In silico methods for vaccine design have gained a foothold over the last decade. Computational approaches for updating vaccines against Influenza have proposed models for predicting antigenic diversity over time, including multivariate regression on physicochemical properties of circulating variants [57], phylogenetic weighting of antigenic evolution [58–60], and dynamic fitness models of antigenic alleles [61]. In silico vaccine design methods have been used for HIV-1 for over two decades to overcome the obstacle posed by HIV-1's extreme diversity [62–64]. There is more diversity within each HIV-1 subtype or CRF than what can be seen across a viral species [65]. Gaschen and colleagues showed that a centralized sequence,

consensus or ancestral, would better represent HIV-1 than any sequence derived from a PLWH [63,66]. To cope with HIV-1 diversity, some designs, such as the mosaic approach, seek to integrate a fraction of the diversity seen in HIV-1 sequences [67–69]. These variability-inclusive strategies are reminiscent of the diversity seen in super-infections, which have previously been associated with the development of neutralization breadth. As such, mosaic antigens are designed to be maximally distant to cover a large fraction of circulating viruses. The rationale is that immunizations with these diverse mosaic inserts, for example corresponding to consensus sequences for group M, subtype B and subtype C, could lead to the development of antibody responses against these distant viruses thereby potentiating broadly cross-reactive responses. Two vaccine efficacy trials are testing the Mosaic design (one reached futility criteria in 2021: https://www.jnj.com/johnson-johnson-and-global-partners-announce-results-from-phase-2b-imbokodo-hiv-vaccine-clinical-trial-in-young-women-in-sub-saharan-africa). An opposite strategy to Mosaic designs was to focus on only the most conserved elements of HIV-1 [70–73]. This stemmed from the realization that variable segments of HIV-1 functioned as decoys eliciting immune responses that were not optimal and that only a small fraction of HIV-1 diversity could be integrated in a vaccine candidate of practical size. Another currently leading strategy, the germline targeting approach, seeks to improve the longitudinal process seen in individuals infected who later developed breadth [74] by reproducing the directional process that leads to breadth in a minority of individuals through using antigens that correspond to stepwise stages of the co-evolution between the virus and the neutralizing response. Our approach is also based on a process seen in natural infections, whereby infections with multiple founder variants were linked to the subsequent development of neutralization breadth. This multi-founder-like design of minimally distant antigens is also akin to the conserved elements vaccine design rationale. We consider that the 'noisification' of a central consensus sequence at target sites for key antibodies will trigger responses to these antibody epitopes and that the toggle between these minimally distant epitopes will promote a desirable affinity maturation process leading to the development of bnAbs. The fact that our vaccine design was derived from the variability seen in multi-founder acute infections suggests that this strategy with a cocktail of minimally distant antigens may be best suited as a priming immunization. Whether subsequent immunizations should consist of the same set of antigens or a subset of them, or one or more distinct antigens, will need to be evaluated with experimental assays.

In summary, our in silico method generates a set of antigens that bear distinct epitopes, but maintain a minimal global distance across Env, constituting a projected formula for increasing the probability of eliciting bNAbs. We hypothesize that this generic approach can serve to design vaccine candidates with enhanced bnAb-eliciting properties for any given sequence. As such, this approach can be used to design cocktail vaccine candidates adapted to any HIV-1 subtypes and circulating recombinant forms. While this model can also be used to design an HIV-1 group M vaccine cocktail, the idea of a successful universal HIV-1 vaccine is far-fetched when considering lessons from the past forty years of HIV-1 vaccine research.

## Materials and Methods

### RV217 participant sequences

We used env sequences that we previously generated via single genome amplification of HIV-1 on plasma samples collected in the first five weeks after HIV-1 diagnosis in acute infection in participants from the RV217 cohort [25–27,29]. All participants were antiretroviral treatment naïve. We included Env sequences from twelve participants with infections with single founders (median = 10, min = 10, max = 28) who developed <35% neutralization breadth and

from six participants with infections with multiple founders who developed >70% neutralization breadth (median = 11, min = 10, max = 13) (**Table 1**) (another participant with multiple founder variants and >70% neutralization breadth was excluded because the development of neutralization breadth occurred following superinfection). Infections with multiple founder variants are illustrated with highlighter plots [34] (**S1 Fig**); we previously reported that these individuals neutralized 74–88% of a 34-virus panel at 435–2115 days post-diagnosis [28]. Sequences belonging to each founder lineage in multi-founder acute infections were used in the training dataset.

## Independent subtype C sequences

Subtype C Env sequences sampled since 2011 were downloaded from the Los Alamos National Laboratory HIV Sequence Database (https://www.hiv.lanl.gov/components/sequence/HIV/search/search.html). Sequences were excluded if the individuals had been vaccinated, if the sequence did not have a complete open reading frame or did not have a sampling year. One sequence was downloaded per individual and sequences were removed if they were non-independent or an outlier. Hypermutated sequences identified with Hypermut 2.0 [75] (using https://github.com/philliplab/hypermutR) were removed with a Fisher's exact test P<0.1. Sequences were de-duplicated at 95% identity.

## Probabilistic model for sequence simulation

A probabilistic model was developed to simulate Env sequences that replicated the variability of infections with multiple founder variants. The model is trained on two Env alignments, corresponding to the major ($F_1$) and minor ($F_2$) founder lineages identified in a given participant (the model can also be trained on a pooled set of individual lineages). A first-order Markov transition probability matrix, $\Theta$, is estimated as described by Le and Gascuel (LG matrix) [39]; in brief, transition rates are directly computed between pairs of amino acids, including transition rates from/to gaps based on an alignment of empirical sequences. We suggest using a large dataset of independent sequences that would correspond to the subtype of the desired target vaccine candidate (i.e., subtype C sequences if the goal is a subtype C vaccine which implies that the seed sequence corresponds to subtype C). For each alignment,

1. The percentage of non-consensus residues was calculated for each alignment site.

2. For each site, $j$, percentage of non-consensus residues, $\pi_j$, an amino acid in the seeding sequence, $k$, and a transition probability, $\Theta_k$, the probability of transitioning from $k$ to $k'$ was written as

$$P\left(k' | k, \pi_j, \Theta_k\right) = \frac{\Theta_k[k]\pi_j}{\Sigma(\Theta_k[-k])}$$

where $P_j(k')$ is calculated for each site $j$. For each simulated sequence, $\pi_j$ is trained on either $F_1$ or $F_2$ and at each site in the sequence, $j$, the new residue is then drawn from a multinomial probability distribution based on $P_j(k')$. The denominator, $\Sigma(\Theta_k[-k])$, is included to force the probability distribution to sum to 1, such that $P_j(k' \neq k) = 1 - P_j(k' = k)$. For each seeding sequence (e.g., a subtype C consensus sequence), whichever model is initially drawn to simulate the initial amino acid in a sequence is consistently applied to generate the following amino acids in that sequence. For the pooled sets, the less diverse major founder lineages were pooled together and the more diverse minor founder lineages constituted a second set. For each

pooled founder lineage alignment, at each site j, $\pi_j$ was estimated as the maximum percentage of non-consensus residues in any individual alignment in that pool (i.e., this retained within-host diversity levels); however, this could be alternatively modeled such that, at each site j, $\pi_j$ was randomly drawn from an individual alignment in the pooled alignment.

## Sequence simulation

One thousand sequences were simulated using a probabilistic model with an empirical transition probability matrix, $\Theta$, computed on an alignment of 172 subtype C Env sequences sampled since 2011 and seeded with a consensus sequence corresponding to sequences from an independent acutely-infected RV217 participant (id = 10066) or a subtype C consensus. Seven sets of sequences were simulated: one trained on each of the six individual training alignments separately and one on a pooled alignment of all of the training sequences. For the pooled model, the $F_1(\pi_j)$ and $F_2(\pi_j)$ were the maximum percentage of non-consensus residues found at site j in a given individual across all $F_1$ and $F_2$ alignments, respectively. The probability of drawing from each variant model was recorded, so the outputs could be analyzed separately.

## Sequence analysis

Consensus sequences were computed with a majority rule. Sequences were aligned to the HXB2 reference in MAFFT v7.419 [76]. For alignments of sequences sampled from participants in RV217, the percentage of non-consensus (as well as non-gap, non-ambiguous) residues at each site was calculated as the number of residues at each site different from the majority consensus residue for that alignment divided by the total number of sequences. Polymorphic sites were defined as sites with at least one amino acid different from the consensus. For simulated sequences, the percentage of non-consensus residues and polymorphic sites were defined against the seeding sequence. Contact sites for known HIV-1 antibodies (n = 116) were previously reported in studies of natural HIV-1 infection (https://www.hiv.lanl. gov/components/sequence/HIV/featuredb/search/env_ab_search_pub.comp).

A maximum-likelihood model of pairwise sequence distance that corrects for sequence length was computed using the *dist.ml* function [77]. Sequence divergence was calculated against the seeding sequence for each alignment. Phylogenies of aligned sequences were constructed with IQ-TREE 2 [78] based on the model with the lowest BIC identified with Model-Finder [79]. The modified graph Laplacian (MGL) is computed for the distance matrix of the reconstructed phylogeny of sequences; eigenvalues calculated from the MGL define the connectivity of the phylogeny in terms of substitutions. Spectral density profile summary statistics represent different aspects of the topology of the phylogeny, such as the longest path through the phylogeny, $\lambda^*$ which is a correlate of non-synonymous/synonymous substitution rates, the proportion of long versus short branching-events, $\psi$, which is a correlate of rate heterogeneity, and the occurrence of branching-events, $\eta$, which is a correlate of transition-transversion rates [40,41]. Spectral density profile summary statistics $\lambda^*,\psi$, and $\eta$ were estimated for phylogenies reconstructed from empirical and simulated sequences. Simulated alignments were iteratively down-sampled 100 times to a random set of 10 sequences. Divergence, pairwise distance, and phylogenetic metrics were calculated on each downsampled alignment.

## Subtype C sequence antigen prediction

A phylogeny for subtype C sequences was constructed with IQ-TREE 2 [78] based on the model with the lowest BIC identified with ModelFinder [79]. Divergence from the majority consensus was computed for each sequence and pairwise distances were computed for all sequences.

For the subtype C consensus, exposed residues (i.e., accessible to antibodies) were defined as we previously described [27] and B-cell epitopes were predicted with Bepi-Pred 2.0 [80] using an epitope prediction threshold of 0.5. The number of polymorphic sites among simulated sequences corresponding to predicted B-cell epitopes were quantified.

To select candidate antigen sequences, simulated sequences were filtered by those that had a non-consensus residue (with respect to the subtype C consensus) in at least three key Ab epitopes (VRC01, CAP256-VRC26.25, PGT121, 10E8, and 35O22). Of these sequences with minimal variability, the two maximally divergent sequences simulated by $F_1$ and two by $F_2$ were selected as candidate antigens. Maximally divergent sequences were selected to cover as much genetic space as possible within the simulated minimal divergence.

## Structure prediction and visualization

The structure of one subunit of the Env-trimer for subtype C consensus (the seed sequence) and four in silico-derived antigenic sequences were predicted with ColabFold [81]. The alignment was prepared using MMseqs2 [82] and the structure prediction was carried out with AlphaFold2 [47]. Before feeding to the ColabFold, the signal peptide and the sequence after the transmembrane helix were removed from the sequence. The structure figure is rendered by PyMol (https://pymol.org/). If a substitution is between a pair of highly similar residues (RK, QE, QN, ED, DN, TS, SA, VI, IL, LM, and FY), the residue is colored yellow; other changes are colored red.

## Statistical analysis

Shapiro's normality test was used to determine if data were normally distributed. If data were normally distributed, pairwise comparisons were made using a Student's t-test; and otherwise using a Mann-Whitney U test. Two-sample Kolmogorov-Smirnov tests were used to compare distributions. Statistical tests were only used to compare empirical data but not simulated data. Comparisons with simulated data were made by assessing inclusion/exclusion of values within the 95% confidence intervals of simulated data.

## Supporting information

**S1 Fig. Highlighter plots of 6 RV217 infections with multiple founder variants.** For each individual, a highlighter plot is constructed from sequences sampled during acute infection using the consensus as the master sequence. The number of days post-diagnosis at which each sequence was sampled is listed to the right of each plot.
(TIF)

**S2 Fig. Phylogenies of 6 RV217 infections with multiple founder variants.** For each individual, a phylogeny constructed from sequences sampled during acute infection and rooted on the majority consensus sequence.
(TIF)

**S3 Fig. Mismatched sites between seeding, training, and simulated sequences.** Correlation plot of the percentage of mismatched non-gapped sites between the consensus of the seeding alignment, simulated alignment, and training alignment for sequences simulated under each RV217 multi-founder infection.
(TIF)

**S1 File. Candidate antigen sequences simulated from a subtype C Env consensus sequence and trained on pooled founder alignments sampled from six multi-founder acute**

**infections in RV217.**
(TXT)

## Acknowledgments

We are indebted to the RV217 participants and clinical team. We also thank Julie Ake, Merlin Robb and Sandhya Vasan.

The views expressed are those of the authors and should not be construed to represent the positions of the U.S. Army, the Department of Defense, or the Department of Health and Human Services.

## Author Contributions

**Conceptualization:** Eric Lewitus, Morgane Rolland.

**Data curation:** Jennifer Hoang, Yifan Li.

**Formal analysis:** Eric Lewitus, Morgane Rolland.

**Investigation:** Eric Lewitus, Yifan Li, Hongjun Bai.

**Methodology:** Eric Lewitus, Yifan Li, Hongjun Bai, Morgane Rolland.

**Resources:** Jennifer Hoang, Yifan Li.

**Software:** Eric Lewitus.

**Visualization:** Eric Lewitus, Hongjun Bai.

**Writing – original draft:** Eric Lewitus, Morgane Rolland.

**Writing – review & editing:** Eric Lewitus, Jennifer Hoang, Yifan Li, Hongjun Bai, Morgane Rolland.

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
