## [Decision Letter · Decision Letter 0]

19 Jun 2022

Dear Dr Rolland,

Thank you very much for submitting your manuscript "Optimal sequence-based design for multi-antigen HIV-1 vaccines using minimally distant antigens" for consideration at PLOS Computational Biology.

As with all papers reviewed by the journal, your manuscript was reviewed by members of the editorial board and by several independent reviewers. In light of the reviews (below this email), we would like to invite the resubmission of a significantly-revised version that takes into account the reviewers' comments.

As you can from both reviews, the methods used in the papers should be much better explained!

We cannot make any decision about publication until we have seen the revised manuscript and your response to the reviewers' comments. Your revised manuscript is also likely to be sent to reviewers for further evaluation.

Sincerely,

Rob J. De Boer

Deputy Editor

PLOS Computational Biology

Reviewer's Responses to Questions

**Comments to the Authors:**

Reviewer #1: Lewitus et al describe a method to simulate HIV antigenic diversity using a hidden Markov model, intended to replicate the diversity found in individuals who produce broadly neutralising antibodies. The intended application is future vaccine design. This is naturally a potentially worthwhile direction of research and a potentially fruitful approach. Unfortunately the presented manuscript has simply too many unanswered questions for me to be able to recommend publication.

Unanswered question #1: What is the actual model here? As a reader unfamiliar with how HMMs are used in this field, the description of how the model works (L458-482) is unsatisfactory. Neither the underlying process nor the observation process are described. Step 1 is simply “a transition rate matrix was estimated”, but how was it estimated, and indeed, what are we transitioning between? Is this a model where evolution happens over time, or between subsequent sites on the genome? Nowhere in the paper is this made clear. The mathematical notation is equally confusing, e.g. “n_k is the number of occurrences” – the number of occurrences of what? Why does theta’, which is (I think) a matrix, occur multiple times as a constant in the multinomial p.m.f.? With j defined as a site and k as an amino acid, how can we ever have j=k? This is entirely incoherent. It makes it impossible for me to properly evaluate the potential strengths and weaknesses of the model. I accept that this may be one of a standard class of HMMs which I am not familiar with – but no other example is cited, and even if this is true it does not excuse the failings of the notation in this manuscript.

Unanswered question #2: What is to be made of the difference in variances in divergence and pairwise distance between the simulated and actual (multiple) sequences in figure 4? I presume that the reported p-values are for t-tests (i.e. equality of means); it seems highly unlikely that p>0.05 would be obtained if testing for equality of variances. It would seem that there is potentially much more diversity in real data than this model can readily capture in simulations. In that case, how generally useful is this?

Unanswered question #3: Is the phylogenetic pattern of one very diverse collection of sequences and one not very diverse collection typical of those samples identified as multiple founder? I would not say, on the face of it, that figures 5A or 6D look like multi-founder infections; in fact it rather looks like viruses in “founder 2” are closer to those in “founder 1” than they are to each other. So what actually links those in “founder 2” to each other? Why does that pattern not suggest that all samples are descendants of a single founder strain with a bimodal distribution of subsequent mutations? Are the authors sure that they have robustly identified infections with multiple founder strains, as the one presented example seems to beg a lot of questions? If the multiple-founder status of individual 40363 is established in another way, the question remains – is this pattern typical?

As a final comment: the use of p-values in a simulation study is always a little unsatisfactory because if a difference actually exists, any desired p threshold can be met by simply doing more simulations.

Reviewer #2: Lewitus et al describe a method for 'evolving' HIV env populations in silico from a seed sequence, where the goal is to simulate the type of within-host diversity that naturally occurs after infection with more than one founder variant. The authors propose that selected sequences from these in silico-evolved populations could be used as (multi-valent) HIV vaccine antigens, to elicit superior antibody neutralization breadth. This vaccine antigen design approach is supported by recent work by the authors, currently under review, showing that individuals with multiple founder variants were more likely to develop neutralization breadth years later. The authors conclude their results section by applying their method to "evolve" a subtype C consensus sequence, and propose that the approach could be applied to any subtype (or recombinant) consensus to generate subtype-specific vaccine antigens.

This is a novel approach that addresses an important health problem, but I have a number of concerns, which, if founded, could considerably reduce the relevance of the approach.

1. Dual-founder infections are initiated by two different (yet still closely related) sequences (as both founders are transmitted from the same donor). But, the authors do not seed their simulations with two different founders. Rather they start with a single seed, and 'evolve' it under two different rates, to produce descendant populations. These rates are based on empirical data from an individual with a dual founder infection (40363), and the approach does yield diverse descendant populations. But, the second rate, which was computed based on the diversity in the sequences that were identified as the descendants of 40363's second founder virus, seems unbelievably high. Is 40363's dataset a really "representative" one to use for this purpose? Perhaps more importantly, is this extremely high rate even accurate? Looking at the partial phylogeny of 40363's sequences (Figure 5A, bottom) I wondered if the sequences that were identified as the descendants of their second founder virus might in fact comprise two populations, namely: the *actual* descendants of the second founder virus (three sequences at the bottom of this tree) and recombinants of the first and second founders (the three divergent sequences in the middle of this tree). This indeed would produce a transition rate matrix with very high the rate. But, evolving sequences this way may not actually yield appropriate vaccine antigens.

This is worth addressing because the authors identify a "bimodal phylogenetic topology" as a key correlate of eliciting bNabs (line 376). Dual founder viruses however will produce a specific type of bimodal tree topology because the two founder sequences are different (as mentioned above there may also be recombination between descendant founders, yielding populations that look "intermediate" in a tree). But the bimodal phylogenies produced by the authors' approach could be of a very different type than 'natural' ones. The partial trees shown in 5A, and the presentation of results as tree summary statistics (where topological information is lost) is insufficient to alleviate my concern. (note: the authors do train the model on founder variants from six other participants, though it is not specified whether these also had widely different rate matrices like 40363's did, and they do present favorable summary statistics, but again no trees are shown).

To summarize, the authors should address:

- why they chose to simulate multiple founder descendant populations by applying different mutation rates to a single founder (rather than applying similar rates to distinct yet closely related founders)

- whether the vastly different rates for 40363's two founder populations are truly representative

- whether the phylogenies inferred from the simulated sequences are topologically similar to sequences sampled from real dual-variant infections.

- the potential implications of recombination in the empirical data, as well as the approach

2. Though the authors indicate that individuals with dual-founder infections are more likely to eventually produce bNAbs, this takes years because these antibodies are produced as a result of ongoing co-evolutionary process where Abs adapt to ever-evolving immune-escape variants in vivo (and some current clinical trials are sequentially immunizing with increasingly "evolved" immunogens to recapitulate this). The authors' approach however appears to design immunogens based on the immediate descendants of the founder viruses only, but does not address the issue of time, nor this biological process of Ab/HIV co-evolution. Or, are the authors proposing that this approach could be used to "evolve" immunogens in silico to generate sequential immunogens? This question ties into my question 3d) below regarding whether this process is really a Markov chain.

3. The paper would benefit from additional explanation of key biological and computational concepts to make it more accessible to readers with diverse expertise. Specifically:

a) Can the authors define what they mean by "minimally distant antigens"? I suspect that lines 408-410 contain the necessary information, but these lines don't come until the paper's conclusion. Adding to this confusion is that the authors, in their evolution of the subtype C strain, specifically select four *maximally diverse* sequences, presumably as their candidate vaccine antigens.

b) The authors should explicitly state, early on, that multi-founder virus infections happen when more than one donor sequence is transmitted to the recipient (and therefore that these viruses, though distinct from one another, are still quite closely related). While this will be obvious to experts, failing to state this could confuse non-experts, particularly as the authors mention superinfection and within- and between- subtype HIV diversity (where the scale of this type of diversity are far, far higher than that seen in a dual- or multi-variant infection).

c) The authors should avoid using the term "founder variant" to refer to BOTH the founder variant and its descendants. Use a different term for the latter. Also, the authors should state explicitly how the founder virus sequence is inferred from its descendant populations (by taking the population consensus of early descendants)

d) It is not clear that the use of "Markov chain" and "hidden Markov model" are used appropriately. Authors state that "sequences are simulated using a Markov chain". From the figure however, it seems that,

a set of output sequence (i.e. s1, s2, s3, etc) are produced by applying the relevant transition matrix (F1 or F2, each with a 0.5 probability) to the seed sequence. Is this correct, or are the output sequences produced stepwise from one another (ie s3 is generated by "mutating" s2, which was generated by "mutating s1")? If all simulated sequences are derived directly from the seed, then the use of "Markov chain" is somewhat misleading (as only the first step of the chain is ever performed, though repeatedly). Similarly, can the authors explain why they refer to the model as a *hidden* Markov model? Specifically, the founder viruses and transition matrices, which would normally be the aspects of the model that would be "hidden", are instead treated as known. Given this, can the authors clarify their use of this terminology?

e) The notation in the equations on p. 22 (step 3) are hard to follow, particularly the definitions of \\Theta and \\Theta'. If j is a site, k is an amino acid, and i_j is *also* a site, what does \\Theta[i_j,] mean (the comma must be a typo), and what does \\Theta'[i_{j=k}] mean? Then, what does \\Theta[i_{j\\noteq k}] mean? We think what you mean is that i_j is the amino acid at site j, though the rest of the notation is still unclear.

Moreover, in step 4, the authors' statement that a new amino acid was "drawn from a multinomial probability distribution" seems unnecessarily complex. In their setting, is N ever anything other than 1, and are n_1, n_2, ..., n_k ever anything other than a tuple of 0s and 1s where exactly one element is 1? It seems that it would be simpler to say that the new amino acid would be drawn according to the transition probability matrix.

A clearer way to rewrite steps 3 and 4 may be to write \\Theta_{k,l} for the transition probability of going from amino acid k to amino acid l (since j is used to denote a site). And, for each site j, make a transition with probability \\pi_j, and stay put with probability 1-\\pi_j. Then, if we have chosen to make a transition, we pick a new state according to the probabilities P(transition from i_j to k) = \\Theta_{i_j, k}.  

 

f) Speaking of \\Theta and \\Theta', is the term "transition rate matrix" used correctly here? As I understand it, this process is a discrete time process so these are "transition probability matrices", not "rate" matrices.

g) Could the authors provide more detail on how the transition matrices are found. Are these directly computed from the empirical distribution of all mutations in each alignment, where the reference sequence is the consensus of the alignment (ie the founder virus)? If so, this should be stated directly.

4. How could this approach be used to develop a universal vaccine for HIV-1 group M? (lines 412-413)?

5. The discussion should feature a discussion of potential caveats/limitations of the approach.

Additional comments:

1) In figure 1E, the notation P(F1 U F2)= 0.5 is confusing. I assume this means that F1 and F2 have an equal (0.5) probability of being applied to a given output sequence. If so, this should be written as P(F1)=0.5; P(F2)=0.5

2) The results (lines 140-142) states that mutations occur... "such that at each iteration, , site of the seeding sequence had a probability of mutating proportional to to a new residue defined by , where and were randomly drawn from either 1 or 2 (Fig 1E).

But, this wording suggests that each *site* in a sequence could be subjected to either the mutation rules of F1 or the mutation rules of F2, independent of the other sites in that sequence. The legend of Figure 1E suggests the same. However, this approach could not produce the output shown in Figure 2. Instead, all sites within a given sequence must be subjected to either the F1 or F2 transition matrix... (?). Please clarify at what level (i.e. sequence or site) the choice of mutation rules is made.

**Have the authors made all data and (if applicable) computational code underlying the findings in their manuscript fully available?**

Reviewer #1: **No: **Code is stated as being to follow but it is not currently available

Reviewer #2: Yes

PLOS authors have the option to publish the peer review history of their article (what does this mean?). If published, this will include your full peer review and any attached files.

Reviewer #1: No

Reviewer #2: No
---

## [Decision Letter · Decision Letter 1]

23 Sep 2022

Dear Dr Rolland,

Thank you very much for submitting your manuscript "Optimal sequence-based design for multi-antigen HIV-1 vaccines using minimally distant antigens" for consideration at PLOS Computational Biology. As with all papers reviewed by the journal, your manuscript was reviewed by members of the editorial board and by several independent reviewers. The reviewers appreciated the attention to an important topic. Based on the reviews, we are likely to accept this manuscript for publication, providing that you modify the manuscript according to the review recommendations.

Sincerely,

Rob J. De Boer

Section Editor

PLOS Computational Biology

[LINK]

Reviewer's Responses to Questions

**Comments to the Authors:**

Reviewer #1: I thank the authors for the extensive work done to respond to the first round of reviews. My remaining comments are minor:

I think that the process by which a transition matrix is estimated from the Los Alamos data is either missing a citation or underdescribed (L150, L429)

On line 164 the number of polymorphisms from training alignments is described as summarised by the mean, but in the legend of figure 2G the text indicates the median was used (L204).

Also in figure 2G legend the "dashed lines" could refer just to the error bars (L204); this is a reference to the horizontal dashed lines presumably.

I don't follow the argument that the bimodal or multimodal phylogenies in figure 4B must reflect recombination. These are simulated sequences and recombination was not explicitly modelled. If the argument is that recombination in the real alignment is the cause of these patterns, or that the simulation process generates patterns that mimic recombination, then this needs fleshing out.

The x-axis is missing from figure 4D.

Reviewer #2: The manuscript by Lewitus et al is substantially improved and many of the concerns raised by reviewers have been addressed. There are still some points that would benefit from clarification, mostly related to the mathematical notation, and some additional minor suggestions to improve the MS, as follows

Related to the mathematical notation and model:

1. line 94-

- One improvement would be to use two separate names for \\pi when it comes from F_1 and F_2. It appears that F_1(\\pi_j) and F_2(\\pi_j) are used to denote this (is this correct?), but it would be better to make this explicit, as this notation is introduced without explanation in lines 157-160

- The authors present the steps starting at the level of a single site, but this later requires some backtracking ("actually all of these sites follow the same template laid out for the entire sequence..." [paraphrasing]). Instead, a better starting point may be "at each iteration, choose one of F_1 and F_2." Everything flows better from there, since the entire sequence is be generated from that choice. (The same holds to some extent for lines 121-123.)

line 110: Is the transition matrix is calculated for overall rate, or for each position? It appears that it is the former, but the notation at line 119 seems to imply that it is per-site, which is confusing.

line 118: This is where the results could say: "At each iteration, choose one of F_1 or F_2. In this example, we will assume F_1 is chosen. Site j of the seeding sequence will have a probability of mutation proportional to F_1(\\pi_j); if it mutates, it will mutate according to the transition matrix \\theta".

line 119-120: Is there any reason to not simply do N/2 iterations with F_1 and then N/2 iterations with F_2? Doing so could also simplify this description.

line 145: perhaps change to "across *the 6* participants" to make it clear that these summary stats are for the 6 participants from RV217 with the multiple founder infections (i.e. those in figure 2)

lines 149-151: Suggestion to move this description of how \\theta is defined to the first mention of \\theta at line 110.

line 287: It is unclear how fig 5a-f show this. There is no scale on the x-axis.

line 439: Is the denominator needed? The normalization should presumably be inherent in \\theta. Also, is \\theta_k[k'] "the probability of a transition from k to k', assuming such a transition happens"? If so, should this be \\theta_k[k'] in the numerator?

line 441: P_n has not been defined anywhere; this is just P, for this step. Perhaps you meant to say "At each step n" in line 434. This would also be a good place to say "first pick F_1 or F_2", or "we first simulate N/2 sequences based on F_1 and then N/2 sequences based on F_2".

Other comments:

1. The manuscript is much more clear, but a brief accessible explanation of why "minimally distant" antigens are optimal for multivalent immunogen design, as early as possible in the introduction, would also be helpful

2. The authors' clarification in the response letter that the approach will primarily produce designs for immunogens used in *priming* vaccinations is helpful - as these are unlikely, on their own, to elicit NAb responses. However, I recommend that this is made more clear in the discussion (e.g. perhaps at the end of line 394 or concluding paragraph) as well as briefly in the abstract and/or author summary.

3. 5/6 of the training alignments are from individuals with CRF02_AE or A1, yet the model was used to produce simulated subtype C sequences (presumably as this the most prevalent subtype worldwide). Some readers may wonder whether this may be a concern; the authors may wish to briefly address this in the discussion.

4. The colors in Figure 2 are re-used in a confusing way. e.g. - maroon is used for simulated data in 2C, D, but the legend of 2F indicates that maroon denotes training data. Similarly, blue is used in 2B, C and D to denote F2, but in 2F it denotes simulated data. MF is also the same blue color, but since this is from the pooled alignment, it may be better to use a different color entirely. It could be helpful to match Figure 3 as well.

5. It would be helpful to have slightly more information on the alignment of 172 subtype C sequences from LANL, as there appear to be far more than this since 2011. Was this restricted to single-genome amplified sequences? HIV RNA (or was DNA also allowed)? What is meant by "sequences were de-duplicated at 95% identity"?

6. recommendation to avoid using "infected" when describing people with HIV, instead use "person with HIV" or "person living with HIV" e.g. lines 42, 65, 368

7. small typos: line 42 comma should be a period. line 372: "breath" should be "breadth".

**Have the authors made all data and (if applicable) computational code underlying the findings in their manuscript fully available?**

Reviewer #1: **No: **As in the previous version, it is stated that code will be provided but it is not currently present

Reviewer #2: Yes

PLOS authors have the option to publish the peer review history of their article (what does this mean?). If published, this will include your full peer review and any attached files.

Reviewer #1: No

Reviewer #2: No

Figure Files:

Data Requirements:

Reproducibility:

References:

---

## [Editor Report · Decision Letter 2]

3 Oct 2022

Dear Dr Rolland,

We are pleased to inform you that your manuscript 'Optimal sequence-based design for multi-antigen HIV-1 vaccines using minimally distant antigens' has been provisionally accepted for publication in PLOS Computational Biology.

Best regards,

Rob J. De Boer

Section Editor

PLOS Computational Biology

Rob De Boer

Section Editor

PLOS Computational Biology

---

## [Editor Report · Acceptance letter]

17 Oct 2022

PCOMPBIOL-D-21-02212R2 

Optimal sequence-based design for multi-antigen HIV-1 vaccines using minimally distant antigens

Dear Dr Rolland,

I am pleased to inform you that your manuscript has been formally accepted for publication in PLOS Computational Biology. Your manuscript is now with our production department and you will be notified of the publication date in due course.

With kind regards,

Anita Estes
